# Identification of Plant-Derived Bioactive Compounds Using Affinity Mass Spectrometry and Molecular Networking

**DOI:** 10.3390/metabo12090863

**Published:** 2022-09-14

**Authors:** Thabo Ramatapa, Anathi Msobo, Pfano W. Maphari, Efficient N. Ncube, Noluyolo Nogemane, Msizi I. Mhlongo

**Affiliations:** 1Research Centre for Plant Metabolomics, Department of Biochemistry, University of Johannesburg, P.O. Box 524, Auckland Park, Johannesburg 2006, South Africa; 2Department of Agriculture and Animal Health, Florida Campus, University of South Africa, Florida 1710, South Africa

**Keywords:** affinity selection-mass spectrometry, drug discovery, high-throughput screening, ligands, natural products, virtual screening, secondary metabolites

## Abstract

Affinity selection-mass spectrometry (AS-MS) is a label-free binding assay system that uses UHPLC-MS size-based separation methods to separate target-compound complexes from unbound compounds, identify bound compounds, classify compound binding sites, quantify the dissociation rate constant of compounds, and characterize affinity-extracted ligands. This label-free binding assay, in contrast to conventional biochemical (i.e., high-throughput screening (HTS)) approaches, is applicable to any drug target, and is also concise, accurate, and adaptable. Although AS-MS is an innovative approach for identifying lead compounds, the possibilities of finding bioactive compounds are limited by competitive binding, which occurs during the equilibration of extracts with the target protein(s). Here, we discuss the potential for metabolite profiling complemented with molecular networking to be used alongside AS-MS to improve the identification of bioactive compounds in plant extracts. AS-MS has gained significant prominence in HTS labs and shows potential to emerge as the driving force behind novel drug development in the future.

## 1. Introduction

Natural products are secondary metabolites produced by organisms, particularly microorganisms and plants. As rich sources of bioactive compounds that are both medically and industrially relevant, natural products are key sources of novel medications and good lead compounds that can be modified extensively during the drug development process, notably as anticancer and antibacterial agents [1,2,3]. Secondary metabolites derived from natural sources are often regarded as having greater “drug-likeness and biological friendliness” than synthetically produced compounds, making them potential candidates for future drug development [4]. In addition to a wide range of scaffold variety and structural complexity [5], natural product extracts have diverse pharmacophores and a great degree of stereochemistry. Consequently, the varied forms and complicated carbon skeletons of natural products have resulted in a substantial fraction of natural products being used in drug discovery [4,6,7]. 

Previously, culture broths or extracts of microorganisms and plants were screened for desired activities, followed by compound isolation, activity screening, and structural analysis. The functional activity of the target is the constraint of most traditional high-throughput screening (HTS) assays, such as enzymatic or G-protein-coupled receptor (GPCR) signaling [8]. Although an effective approach to finding natural products with desired activities, the increased rediscovery rate of the same compounds following the tedious screening, purification, and identification stages make the discovery of new natural products increasingly difficult. Furthermore, each stage is time-consuming, labor-intensive, and, more importantly, compound-untargeted. As a result, developing an efficient and selective screening strategy for active compounds is essential.

## 2. Affinity Selection-Mass Spectrometry-Based Drug Discovery versus the Conventional Method

Recent advances in genomics technologies have revealed a wealth of natural compounds that are yet to be found and defined [3,5,9,10]. To identify new lead compounds for receptors that are known or suspected to be involved in a disease pathway, target-based screening approaches are performed [11]. In this regard, affinity selection-mass spectrometry (AS-MS), allows for the simultaneous characterization and dereplication of active ingredients in complex mixtures, such as extracts of botanicals, fungi, and microbial cultures [3]. AS-MS is a label-free binding assay system that uses UHPLC-MS size-based separation methods, such as ultrafiltration, gel permeation, or size-exclusion chromatography, to separate target-compound complexes from unbound compounds, identify bound compounds, classify compound binding sites, quantify the dissociation rate constant of compounds, and characterize affinity-extracted ligands [3,11,12,13]. The AS-MS method entails the expression of target proteins which are later immobilized onto the chromatographic column, followed by the equilibration of target proteins with the crude extract. This leaves bioactive compounds bound to the target protein, which are then characterized using LC-MS (Figure 1). Different types of AS-MS have been used to identify bioactive compounds from plant extracts, including pulsed ultrafiltration (PUF) AS-MS, size-exclusion chromatography (SEC) AS-MS, and magnetic microbead affinity selection screening (MagMASS) [14,15]. Table 1 summarizes native and affinity MS methods used for the identification of bioactive compounds in plants. 

AS-MS can be either direct, in which the protein–ligand complexes are measured by MS, or indirect, in which the occurrence of a complex is inferred by detecting the ligand after it has dissociated from the protein target [18]. These label-free binding assay approaches are ideal for identifying ligands to target receptors because of their speed, selectivity, and sensitivity. As such, AS-MS allows for the rapid isolation of pharmacologically active molecules from complex mixtures for mass spectrometric characterization and identification [3,19,20,21]. During the AS-MS incubation of targets at a precise stage, the drug targets are frequently present in molar excess relative to probable ligands. This adds to the merits of AS-MS over conventional methods as it reduces competition, which could lead to the detection of ligands with lower affinity for the target [21]. Studies utilizing AS-MS to identify new bioactive metabolites from plant extracts are presented in Table 2 and their main findings are summarized. Furthermore, Table 2 shows docking studies conducted on AS-MS isolated compounds to confirm their interaction with targeted proteins. Molecular docking predicts ligand-target complex’s preferred confirmation and strength of association or binding affinity [16]. 

In comparison to conventional drug discovery (Figure 1), AS-MS offers greater advantages. This label-free binding assay, in contrast to typical biochemical HTS approaches, is applicable to any drug target and is concise, accurate, and adaptable without the requirement for compound or target modification [3,20,21,27]. As a result, AS-MS does not require radiolabels, UV, or fluorescent chromophores, and is compatible with all receptors, enzymes, incubation buffers, cofactors, and ligands [20]. Whereas conventional HTS is designed to discover only orthosteric ligands, AS-MS can also detect allosteric ligands. Moreover, natural product screens generally contain a library of natural extracts, which may not be amenable to conventional target-based assays [27,28,29]. As a result, AS-MS provides access to molecules with far greater chemical diversity than synthetic chemical libraries and can provide direct confirmation of the protein–ligand complex’s presence and stoichiometric information [3,21]. 

### Metabolite Profiling Meeting AS-MS

Although AS-MS is innovative in identifying lead compounds, competitive binding occurs during the equilibration of extracts with the target protein(s) where only the compounds with high binding affinity could be identified as potential leads in drug discovery. However, this phenomenon limits the possibility of finding a wider range of bioactive compounds. The molecular networking (MN) approach has emerged as a useful tool to analyze tandem mass spectrometry (MS2) data to further identify compounds in MS-based metabolite profiling methods. The MN concept is based on organizing and visualizing tandem MS data using a spectral similarity map to identify the presence of homologous MS2 fragmentations. The nodes of structurally related compounds tend to cluster and generate clusters of analogues since they share similar fragmentation spectra [30,31,32]. Thus, mapping AS-MS isolated compounds over a molecular network of different chemical clusters will allow efficient selection of chemical clusters of potential bioactive compounds against a specific protein target (Figure 1). Furthermore, merging AS-MS isolated compounds over a molecular network can help in compound annotation/structural elucidation and activity relationships within a specific chemical cluster. Such compounds can be studied further to explore the binding affinity to the target protein using virtual screening (Figure 1). Virtual screening evaluates vast libraries of chemical compounds using computer technology and software to identify potential drug candidates based on biological structures [33]. The primary purpose of virtual screening is to condense the vast virtual chemical space of small organic molecules that can be synthesized or screened against a single target protein down to a manageable number of compounds that have the best probability of becoming a therapeutic candidate [32,33]. 

For example, in research conducted by Wang and colleagues [24], molecular networking and virtual screening coupled with AS-MS was used to discover two compounds, kurarinol and kurarinone, which were successfully identified from 11 traditional Chinese medicine (TCM) herbs and were confirmed to interact with GTPase of Ras. Initially, Ras protein was expressed; then, affinity mass selection was employed to find compounds active against the GTPase from crude extracts of TCM crude extracts. Thereafter, molecular networking was used to identify the two compounds, kurarinol and kurarinone, in the extracts confirmed to interact with Ras protein by virtual screening [24]. 

## 3. Concluding Remarks

AS-MS approaches can be used investigate the binding of candidate compounds to immobilized targets and serve as supplements to traditional drug development methods. These methods rely on target-based screening strategies to discover alternative lead compounds for receptors that are known or possibly involved in a disease pathway. As the first step in drug discovery, affinity selection-mass spectrometry (AS-MS) is useful for identifying lead compounds in plant extracts. The progress of instrumentation automation, particularly of LC and MS components, would naturally allow for more diverse and innovative applications, potentially expanding the already significant impact of AS-MS on drug discovery. As a result, AS-MS has gained significant prominence in high-throughput screening labs and may be the driving force behind novel small molecule medication development, and it is expected that these applications will continue and expand in the future. Furthermore, mapping of AS-MS isolated bioactive compounds over molecular networks will allow efficient selection of clusters of potential bioactive compounds. The mapping of AS-MS bioactive compounds on molecular networks also offers additional advantages, such as structural elucidation and activity relativity relationships within a chemical cluster to which the isolated compounds belong. 

## Figures and Tables

**Figure 1 metabolites-12-00863-f001:**
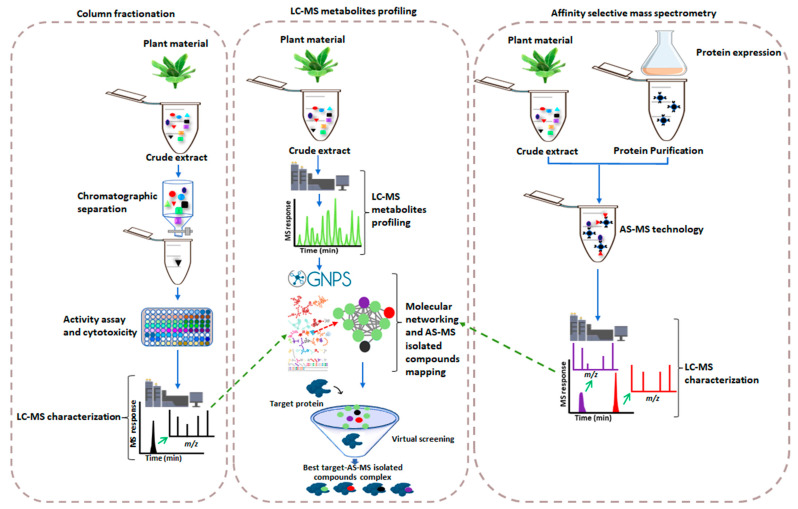
Comparative illustration of conventional drug discovery (on the left) vs. AS-MS-based drug discovery (on the right) and how AS-MS is explored further to increase the possibilities of finding bioactive compounds using molecular networking (in the middle). In the molecular network, compounds that were initially identified using AS-MS (colored red and purple) will appear as nodes with other surrounding nodes (colored green) representing compounds with similar structures; these compounds can be virtually screened for binding affinity to target proteins, increasing the chances of finding more drug lead compounds. Green arrows indicate mapping of the LC-MS characterized ligands into molecular networking.

**Table 1 metabolites-12-00863-t001:** Short description of native and affinity MS methods.

	Description	References
1. Affinity selection		
Magnetic microbead affinity selection screening (MagMASS)	MagMASS is a solid-phase alternative that complements the solution-phase screening approaches. MagMASS involves tethering the target to magnetic microbeads, incubating the immobilized protein with a natural product mixture, using magnetism to separate the ligand-protein/bead complexes from unbound compounds, and then releasing the bound ligands for UHPLC-MS analysis.	[16]
Pulsed ultrafiltration (PUF) AS-MS.	PUF AS-MS screening begins with the incubation of a mixture of compounds, such as a natural product extract with a solution-phase macromolecular receptor (protein, enzyme, or RNA). After equilibrium is achieved, ultrafiltration is used to separate the large ligand-receptor complexes from the unbound low-mass compounds. Because large pore sizes enable faster ultrafiltration separation, the pore size of the ultrafiltration membrane should be as large as possible while still retaining the macromolecular receptor.	[16]
Collision-induced affinity selection mass spectrometry (CIAS-MS)	Collision-induced affinity selection mass spectrometry (CIAS-MS) is a new method that relies on the affinity between a protein and its ligand for the identification of ligands.	[15]
Size exclusion chromatography (SEC) AS-MS	SEC AS-MS is a solution-phase screening approach like PUF AS-MS that begins with the incubation of a mixture of possible ligands with a macromolecular receptor. After equilibrium is achieved, SEC is used to separate the large ligand-receptor complexes from smaller, unbound compounds. The high mass complexes elute first during SEC and are then denatured using an organic solvent to release the ligands for reversed-phase LC-MS analysis.	[16]
2. Native MS		
Bioassay-guided fractionation-MS	This involves the analysis and characterization of molecules whereby the native structural features of the analytes are retained as much as possible. It provides binding informationabout each compound towards the protein of interest.	[17]

**Table 2 metabolites-12-00863-t002:** Identification of plant-derived bioactive compounds using AS-MS.

Plant	Compound	Target	Docking	Main Results	References
*Cannabis sativa*	– Cannabigerolic acid – Cannabidiolic acid – Tetrahydrocannabinolic acid	– COVID-19 Spike protein	Cannabigerolic acid binds to the anallosteric site of S1 with −6.6 kcal/mol binding energy.Cannabidiolic acid also binds at the orthosteric site with −6.3 kcal/mol.THCA-A bind at the orthosteric site with −6.5 kcal/mol binding energy.	Bound to the spike protein thus preventing entry into the cell.	[16]
*Radix salvia miltiorrhiza*	– Salvianolic acid C – Salvianolic acid A	– Xanthine oxidase (XOD)	No	Salvianolic acid C exhibited potent XOD inhibitory activity with an IC_50_ of 9.07 μM.	[22]
*Scutellaria baicalensis*	– Baicalein – Scutellarein – Ganhuangenin	– 3C-like protease (3CLpro)	No	Three flavonoids were identified as potential noncovalent inhibitors against 3CLpro with IC50 values of 0.94, 3.02, and 0.84 µM, respectively.	[21]
*Gancao (licorice root)*	– 18β-glycyrrhetinic acid – Licochalcone A	– Ebola virus (EBOV) nucleoprotein – Marburg virus (MARV) nucleoprotein	In silico docking analysis was employed to create a potential model for binding of GC7 and GC13 to EBOV nucleoprotein.	By combining affinity mass spectrometry and metabolomics approaches, two compounds were identified from a traditional Chinese medicine *Gancao* (licorice root) that binds to nucleoproteins (NPs). These two ligands, 18β-glycyrrhetinic acid, and licochalcone A were verified by defined compound mixture screens and further characterized with individual ligand binding assays.	[23]
*Rhizoma atractylodis macrocephalae* *Rhizoma pinelliae* *Bulbus fritillaria* *Rhizoma paridis* *Rhizoma curcumae* *Fructus trichosanthis* *Rhizoma dioscoreae bulbiferae* *Radix sophorae flavescentis* *Radix ginseng* *Radix notoginseng* *Radix asparagi*	– Kurarinol – Kurarinone	– Ras protein	For the docking analysis, ligands kurarinol, kurarinone, 20(s)-Rg3, and 20(s)-Rh2 were inserted into the GTP-binding pocket and the results demonstrated that kurarinol and kurarinone competed with GTP.	Molecular networking and virtual screening coupled with affinity selection-mass spectrometry discovered two compounds, kurarinol and kurarinone, were confirmed to interact with GTPase of Ras and were successfully identified from 11 traditional Chinese medicine (TCM) herbs.	[24]
*Piper kadsura* *Piper nigrum* *Ophiopogon japonicus* *Salvia miltiorrhiza*	– HJ-1 – HJ-4 – HJ-6	– EBOV nucleoprotein	Molecular docking studies were performed to create a docking model of HJ-4 interacting with the hydrophobic pocket in the C-lobe of the nucleoprotein.	Through affinity selection-mass spectrometry approach, three compounds isolated from *Piper nigrum* (HJ-1, HJ-4, and HJ-6) strongly promoted the formation of large nucleoprotein oligomers and reduced the protein thermal stability, and docking studies were performed to show the interaction of HJ-4 to EBOV nucleoprotein.	[25]
*Glycyrrhiza inflata*	– Licochalcone A	– Spike protein	No	Small molecule ligands to the spike protein were discovered in extracts of the licorice species, *Glycyrrhiza inflata*. In particular, two hits were detected during screening of *Glycyrrhiza inflata,* and hit one was identified as licochalcone A while hit 2 corresponded to licoflavone B and glyinflanin K. However, in the absence of authentic standards, the conformation of this ligand (hit 2) is still ongoing.	[3]
*Rabdosia rubescens*	Oridonin	Nsp9 Protein	No	A known SARS-CoV-2 Nsp9 ligand, oridonin, was successfully detected when it was mixed with Nsp9	[15]
*Tang-zhi-qing*	– 2,3,4,6-Tetra-O-galloyl-d-glucose – 1,2,3,4-Tetra-O-galloyl-d-glucose – 1,2,3,4,6-Penta-O-galloyl-d-glucose – Quercetin-3-O-β-d-glucuronide – Quercetin-3-O-β-d-glucoside	– Maltase – Invertase – Lipase	No	Through the use of multiple target-immobilized magnetic beads coupled with high-performance liquid chromatography–mass spectrometry, five active compounds, namely, 2,3,4,6-tetra-O-galloyl-d-glucose, 1,2,3,4-tetra-O-galloyl-d-glucose, 1,2,3,4,6-penta-O-galloyl-d-glucose, quercetin-3-O-β-d-glucuronide, and quercetin-3-O-β-d-glucoside, were identified and their activities were validated by conventional inhibitory assay.	[26]

## Data Availability

No new data were created or analyzed in this study. Data sharing is not applicable to this article.

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
