# Peer review of "Identification of Plant-Derived Bioactive Compounds Using Affinity Mass Spectrometry and Molecular Networking"

_metabolites, 2022, doi:10.3390/metabo12090863_

Round 1
Reviewer 1 Report
The authors submitted a Mini-Review on Affinity Selection Mass Spectrometry in drug lead discovery. AS-MS and the integration of other bioinformatics tools such as molecular networking in the AS-MS workflow is certainly an interesting research area that warrants a full literature review, which would be of great interest to the scientific community. The way the authors presented the data in the manuscript is rather limited and I would wish for a more complete article. Even though this is a “Mini-review”, it is questionable if 25 references on the topic are adequate and it is not apparent how the selection of references was made. As Metabolites does not provide specific guidelines for a “Mini-review” on their website, it will be up to the editors to decide if the presented Mini-review provides enough detail for publication.
In the abstract the authors state: ”Here we discuss the potential of how untargeted mass spectrometry analysis complemented with molecular networking can be used alongside AS-MS to improve ligand identification in drug discovery studies.” This indicates that the core of the manuscript is the discussion of molecular networking complementing AS-MS. The actual discussion on this topic is however limited to lines 113-121 and the potential and the “how” doesn’t become fully clear and should be described in much more detail in the manuscript.
It should be specified in the abstract and introduction that this review only focuses on examples of plant natural products.
Furthermore, the differences of native MS and affinity selection MS should be discussed as Table 1 includes examples from both approaches.
There a several specific comments that should be addressed:
Line 31: What do the authors mean by “…lead compounds that can be modified extensively during the drug development process”?
Lines 47-48: “Nonetheless, recent advances in genomics technologies have revealed a wealth of natural compounds that have yet to be found and defined” - this sentence appears out of context.
Line 70: Table 1 should have some kind of introduction in the text and the main highlights of the table should also be described in text. Furthermore, Table 1 is not very reader friendly as it occupies too much space.
Lines 82-84: The authors should clarify what they mean by “AS-MS provides access to molecules with far greater chemical diversity than synthetic chemical libraries”
Figure 1: The authors should possibly with lines/boxes/highlights/or different labelling show which part of the figure refers to conventional drug discovery, AS-MS etc
Title of Figure 1: Is there a formatting error or is all the information from lines 87-107 meant to be part of the figure title? If it is in fact part of the title, the formatting should be different.
Lines 116-119: The example given (reference [22]) should be described in more detail; e.g. include the actual name of the metabolite, etc. Is this a hypothesis that the authors established? It does not come clear why this example was chosen and why other studies e.g ref [16] from table 1 that actually used molecular networking are used to demonstrate the utility of the approach.
Formatting of the manuscript needs to be carefully revised, there are many mistakes in capitalisation, spelling, spacing, font formatting, page numbering and references.
Author Response
Comments and Suggestions for Authors:
The authors submitted a Mini-Review on Affinity Selection Mass Spectrometry in drug lead discovery. AS-MS and the integration of other bioinformatics tools such as molecular networking in the AS-MS workflow is certainly an interesting research area that warrants a full literature review, which would be of great interest to the scientific community. The way the authors presented the data in the manuscript is rather limited and I would wish for a more complete article. Even though this is a “Mini-review”, it is questionable if 25 references on the topic are adequate and it is not apparent how the selection of references was made. As Metabolites does not provide specific guidelines for a “Mini-review” on their website, it will be up to the editors to decide if the presented Mini-review provides enough detail for publication.
≥ Thank you very much for your detail feedback on our manuscript.
In the abstract the authors state: ”Here we discuss the potential of how untargeted mass spectrometry analysis complemented with molecular networking can be used alongside AS-MS to improve ligand identification in drug discovery studies.” This indicates that the core of the manuscript is the discussion of molecular networking complementing AS-MS. The actual discussion on this topic is however limited to lines 113-121 and the potential and the “how” doesn’t become fully clear and should be described in much more detail in the manuscript.
≥ The description was made see section 2 (line 103, along with a practical example - line 121
It should be specified in the abstract and introduction that this review only focuses on examples of natural plant products.
≥ The suggestion was made see line 1 (on the title) and line 21 (on the abstract)
Furthermore, the differences of native MS and affinity selection MS should be discussed as Table 1 includes examples from both approaches.
≥ Native and affinity MS methods have been discussed in Table1
There a several specific comments that should be addressed:
Line 31: What do the authors mean by “…lead compounds that can be modified extensively during the drug development process”?
≥ Sentence changed, see line 29
Lines 47-48: “Nonetheless, recent advances in genomics technologies have revealed a wealth of natural compounds that have yet to be found and defined” - this sentence appears out of context.
≥ Sentence moved to line 50-51
Line 70: Table 1 should have some kind of introduction in the text and the main highlights of the table should also be described in text. Furthermore, Table 1 is not very reader friendly as it occupies too much space.
≥ The aim of this table is to summarize AS-MS studies and their finding. Since this is a mini review, we would like to keep the table as it is.
Lines 82-84: The authors should clarify what they mean by “AS-MS provides access to molecules with far greater chemical diversity than synthetic chemical libraries”
≥ Sentence changed, see line 91-93
Figure 1: The authors should possibly with lines/boxes/highlights/or different labelling show which part of the figure refers to conventional drug discovery, AS-MS etc
≥ Figure has been modified to highlight the different methods. See figure 1.
Title of Figure 1: Is there a formatting error or is all the information from lines 87-107 meant to be part of the figure title? If it is in fact part of the title, the formatting should be different.
≥ Title of Figure 1 has been modified and formatted correctly, see line 95- 101
Lines 116-119: The example given (reference [22]) should be described in more detail; e.g. include the actual name of the metabolite, etc. Is this a hypothesis that the authors established? It does not come clear why this example was chosen and why other studies e.g ref [16] from table 1 that actually used molecular networking are used to demonstrate the utility of the approach.
≥ A practical example has been provided as suggested, see line 121-128
Formatting of the manuscript needs to be carefully revised, there are many mistakes in capitalisation, spelling, spacing, font formatting, page numbering and references.
≥ Manuscript was sent for gramma check and formatting. Page and reference number was corrected.
Reviewer 2 Report
The work submitted for review entitled "Affinity selection-mass spectrometry: The game changer in drug lead discovery" is a literature review on drug discovery using AS-MS. The manuscript takes a closer look at what the AS-MS technique is, compares it with conventional methods and gives examples of applications. The paper contains some deficiencies that must be corrected before it is accepted for publication.
1. What is the novelty of the work? Among the cited articles, only 10 were published in the last 5 years. In addition, much of the information in the manuscript can be found in other articles (doi: 10.1038/s41570-020-00229-2; 10.1016/j.cbpa.2007.07.011; 10.1002/jms.4647).
2. The literature review was done quite superficially. 25 references even for a mini-review is a small amount.
3. The second chapter should be rewritten. Examples of AS-MS applications should be moved to the next chapter with a breakdown of proteins and metabolites, while only a comparison with conventional methods should remain in chapter two.
4. In concluding remarks, the authors state that the AS-MS method is useful in identifying chemical compounds from botanical, fungal, and microbial extracts, unfortunately, this is not supported by the text. Table 1 should be supplemented with bacteria and fungi.
5. The description under figure 1 is far too long. Part of the description should be moved to the main text, and the description should be concise.
6. The resolution of figure 1 is insufficient.
7. What is a "Mimi-review"? (First page, upper left corner). It should be "mini-review".
Author Response
Comments and Suggestions for Authors:
The work submitted for review entitled "Affinity selection-mass spectrometry: The game changer in drug lead discovery" is a literature review on drug discovery using AS-MS. The manuscript takes a closer look at what the AS-MS technique is, compares it with conventional methods and gives examples of applications. The paper contains some deficiencies that must be corrected before it is accepted for publication.
≥ Thank you very much for your detail feedback on our manuscript. In this review we discuss how untargeted metabolites profiling complemented with molecular networking can be used to identify ligand that AS-MS could have missed. One of the limitations of AS-MS is competitive binding. Here, only compounds or ligands with a strong affinity to target/protein are identified and compounds with moderate to weaker interactions can be missed. Thus, these compounds can be identified by using untargeted metabolite profiling, molecular docking, and molecular networking, as shown in figure 1.
- What is the novelty of the work? Among the cited articles, only 10 were published in the last 5 years. In addition, much of the information in the manuscript can be found in other articles (doi: 1038/s41570-020-00229-2; 10.1016/j.cbpa.2007.07.011; 10.1002/jms.4647).
≥ As we mentioned earlier on, here we discuss the potential of untargeted metabolomics and molecular networking in complimenting AS-MS. AS-MS experiments have lead to the identification of one or more ligands. However, some ligands could be missed due to AS-MS limitations. Studies have shown that AS-MS isolated ligands belong to the same compound class/group. This suggests that some of the compounds (with strong to moderate interac-tion) could be from the same class. Molecular networking uses spectral data to group compounds into compounds classes; thus, compounds that group with AS-MS isolated ligand could be potential ligand to the targeted.
- The literature review was done quite superficially. 25 references even for a mini-review is a small amount.
≥ More references were added. See reference list (references 7, 8, 9, 12, 13,14, 15 and 27 were added).
- The second chapter should be rewritten. Examples of AS-MS applications should be moved to the next chapter with a breakdown of proteins and metabolites, while only a comparison with conventional methods should remain in chapter two.
≥ Section was moved as suggested. See section 2 (Affinity selection-Mass spectrometry-based drug discovery versus the conventional method)
- In concluding remarks, the authors state that the AS-MS method is useful in identifying chemical compounds from botanical, fungal, and microbial extracts, unfortunately, this is not supported by the text. Table 1 should be supplemented with bacteria and fungi.
≥ The manuscript focus was narrowed to plant-derived ligands or extract, see line 1 (on the title) and line 21 (on the abstract)
- The description under figure 1 is far too long. Part of the description should be moved to the main text, and the description should be concise.
≥ Figure legend shortened and some parts moved to text.
- The resolution of figure 1 is insufficient.
≥ Figure resolution was improved and this is uploaded as a separate file.
- What is a "Mimi-review"? (First page, upper left corner). It should be "mini-review".
≥ Typo corrected
Round 2
Reviewer 1 Report
Through the changes made by the authors the manuscript has noticeably improved and should be considered for publication after a few more changes are made.
1. Line 20: remove “As a result,”
2. I still have an issue with the definition provided for “Native MS” and don’t agree with [18] being an appropriate reference. Please see for example this review: https://doi.org/10.1021/acs.chemrev.1c00212.Or this recent paper https://doi.org/10.1038/s41467-022-32016-6 has provided an easy summary in its introduction.
3. Line 74: it is not clear if “this approach” refers to direct or indirect AS-MS discussed in the sentence before.
4. Line 84 should be something like: Studies utilising AS-MS to identify new bioactive metabolites from plant extracts are presented in Table 2 and their main findings are summarised.
It should further be explained why the docking column was included as the concept was not mentioned in in the text before.
5. Line 111: I would change “process” to “analyse”
6. Line 116-117. The statement cross-references Figure 1 however Figure 1 does not show virtual screening it only briefly mentions it in the figure caption.
7. Line 127: remove 1 “to” before find
8. Line 143: compounds
9. Lines 144-147: I think this is an important statement that should also be discussed in section 2.1.
10. The references still need to be changed to MDPI format.
Author Response
Comments and Suggestions for Authors
Through the changes made by the authors the manuscript has noticeably improved and should be considered for publication after a few more changes are made.
≥ Thank you very much for your detail feedback on our manuscript.
- Line 20: remove “As a result,”
≥ The phrase has been removed see line 20
- I still have an issue with the definition provided for “Native MS” and don’t agree with [18] being an appropriate reference. Please see for example this review: https://doi.org/10.1021/acs.chemrev.1c00212.Or this recent paper https://doi.org/10.1038/s41467-022-32016-6 has provided an easy summary in its introduction.
≥ Thank you very much for sharing these references. Native MS has been defined based on the definition from these references and these were cited.
- Line 74: it is not clear if “this approach” refers to direct or indirect AS-MS discussed in the sentence before.
≥ Here we are referring to direct and indirect AS-MS. This has been corrected see line 74.
- Line 84 should be something like: Studies utilising AS-MS to identify new bioactive metabolites from plant extracts are presented in Table 2 and their main findings are summarised.
It should further be explained why the docking column was included as the concept was not mentioned in in the text before.
≥ Line changed as suggested and molecular docking column has been explained.
- Line 111: I would change “process” to “analyse”
≥ Process changed to analayse. See line 112
- Line 116-117. The statement cross-references Figure 1 however Figure 1 does not show virtual screening it only briefly mentions it in the figure caption.
≥ The figure was modified to include virtual screening
- Line 127: remove 1 “to” before find
≥ “to” was removed. See line 133
- Line 143: compounds
≥ Error has been corrected see line 149
- Lines 144-147: I think this is an important statement that should also be discussed in section 2.1.
≥ The statement has been incorporated to section 2.1 see line 117 - 121
- The references still need to be changed to MDPI format.
≥ References were checked and formatted according to the MDPI reference style.
Reviewer 2 Report
The authors have complied with the reviewers' suggestions. The paper can be published in its present form.
Author Response
Comments and Suggestions for Authors
The authors have complied with the reviewers' suggestions. The paper can be published in its present form.
≥ Thank you very much for reviewing and accepting our manuscript for publication.